Identifying hub genes of papillary thyroid carcinoma in the TCGA and GEO database using bioinformatics analysis

Wan Ying emilywan01@tom.com 1
Zhang Xiaolian 2
Leng Huilin 3
Yin Weihua 4
Zeng Wenxing 1
Zhang Congling 1
1 Department of Inspection, People’s Hospital of Yichun City , Yichun , China
2 Department of Blood Transfusion, People’s Hospital of Yichun City , Yichun , China
3 Department of Neurology, People’s Hospital of Yichun City , Yichun , China
4 Department of Oncology, People’s Hospital of Yichun City , Yichun , China
Uversky Vladimir
Electronic publication date: 2020 Jul 9
Publication date: 2020
Volume: 8
Electronic Location ID: e9120
Received 2019 Nov 25; Accepted 2020 Apr 13
Copyright: ©2020 Wan et al.
Copyright year: 2020
Copyright holder: Wan et al.
License: This is an open access article distributed under the terms of the Creative Commons Attribution License, which permits unrestricted use, distribution, reproduction and adaptation in any medium and for any purpose provided that it is properly attributed. For attribution, the original author(s), title, publication source (PeerJ) and either DOI or URL of the article must be cited.
License URL: https://creativecommons.org/licenses/by/4.0/

Keywords: Thyroid carcinoma, TCGA, GEO, Signature, Prognosis, Biomarker

Funding: The authors received no funding for this work.

==============================
Background

Thyroid carcinoma (THCA) is a common endocrine malignant tumor. Papillary carcinoma with low degree of malignancy and good prognosis is the most common. It can occur at any age, but it is more common in young adults. Although the mortality rate is decreased due to early diagnosis, the survival rate varies depending on the type of tumor. Therefore, the purpose of this study is to identify hub biomarkers and novel therapeutic targets for THCA.

Methods

The GSE3467, GSE3678, GSE33630 and GSE53157 were obtained from the GEO database, including 100 thyroid tumors and 64 normal tissues to obtain the intersection of differentially expressed genes, and a protein-protein interaction network was constructed to obtain the HUB gene. The corresponding overall survival information from The Cancer Genome Atlas Project-THCA was then included in this research. The signature mechanism was studied by analyzing the gene ontology and the Kyoto Encyclopedia of Genes and Genome database.

Results

In this research, we identified eight candidate genes (FN1, CCND1, CDH2, CXCL12, MET, IRS1, DCN and FMOD) from the network. Also, expression verification and survival analysis of these candidate genes based on the TCGA database indicate the robustness of the above results. Finally, our hospital samples validated the expression levels of these genes.

Conclusion

The research identified eight mRNA (four up–regulated and four down–regulated) which serve as signatures and could be a potential prognostic marker of THCA.

Introduction

Thyroid carcinoma (THCA) originates from thyroid tissue and has the potential for distal metastasis (Siegel, Miller & Jemal, 2019). It is worth noting that the incidence in women is two to four times higher than that in men (Carling & Udelsman, 2014). THCA is mainly divided into four categories, including papillary carcinoma (85%), follicular carcinoma (10–15%), medullary carcinoma (5–10%) and undifferentiated THCA (<5%). As a result of early diagnosis, the mortality rate of THCA is stable or declining, although the incidence of THCA continues to rise (Wang et al., 2019b). Notwithstanding, the high incidence of papillary thyroid cancer (PTC) depends on many uncontrollable factors, such as specific genetic changes of THCA, and traumatic invasive examination has different degrees of damage to the patient’s thyroid. Therefore, it is very important to identify hub biomarkers and non-invasive methods for PTC.

With the development of RNA sequencing and microarray technology, the study of differentially expressed genes (DEGs) between cancer and normal tissues has improved our understanding of the molecular mechanism of PTC. It also could help us identify important biomarkers. Previous studies were based on one dataset with limited sample size, and this is not the best way to get DEGs from a single dataset (Liang & Sun, 2018; Zhao et al., 2016). The above defects may lead to a deviation in the final result.

In this research, we conducted a multi-step analysis through comprehensive bioinformatics analysis to identify hub genes in THCA. We obtained the expression profiles of GSE3467, GSE3678, GSE33630 and GSE53157 with PTC from the GEO database, which included a total of 100 THCA and 64 normal samples. After the intersection, we got a total of 179 DEGs in four datasets. Then, we carried out bioinformatics analysis on these genes. This analysis provided reliable and novel biomarkers for THCA, which will be helpful for further clinical application in the diagnosis, prognosis and targeted treatment of PTC.

Methods

Datasets selecting and DEGs identification

We downloaded the four gene expression datasets of PTC from GEO database, including the following criterias: (a) PTC, (b) datasets including tumor and normal tissues, (c) the organism is Homo sapiens, (d) Sample size exceeding 10 samples. GSE3467 (n = 18), GSE3678 (n = 14), GSE33630 (n = 60) and GSE53157 (n = 27) were among depended on the GPL570 platform. The limma package was used to identify the DEGs in each GEO datasets in R (Version 3.7.1) (Ritchie et al., 2015). The P-value is determined by the false discovery rate. The standard for DEG is that the false discovery rate is less than 0.05, and the criteria of the groups were |log2FC(fold change)| ≥ 1.

KEGG and GO enrichment analyses of DEGs

To explore biological information and obtain more comprehensive gene and protein functions, we used clusterProfile packages for GO, which including cellular component (CC) biological process (BP) and molecular function (MF), and KEGG analysis (Yu et al., 2012).

PPI network construction

The online database is used to predict the PPI network, which is used to retrieve the interacting genes (STRING; http://string-db.org) (Szklarczyk et al., 2017). In this research, the PPI network of genes was established with STRING database, and the interaction with high confidence >0.4 was statistically significant. Next, we use Cytoscape (Version 3.6.1), a free bioinformatics software, to visualise and analyse the molecular interaction networks.

TCGA database validation and survival analysis

In order to verify the results of the hub genes, we also obtained the gene expression profiles from The Cancer Genome Atlas (TCGA) (https://cancergenome.nih.gov/) database to validate them, including 502 tumor and 58 non-tumor samples. SurvExpress, a free online website, uses log-rank tests to draw Kaplan–Meier plots that contain the “Survival” package of R software.

Clinical patient samples

A total of 39 patients with THCA diagnosed in People’s Hospital of Yichun City from October 2018 to August 2019 were included in this study. This study was approved by the Ethical Committee of People’s Hospital of Yichun City (PHYC-2018-10-8-1). All patients had not received any radiotherapy or chemotherapy before sampling and signed a written informed consent form. Immediately after surgical resection, the tumor and adjacent non-tumor tissues were transferred to liquid nitrogen and stored in the refrigerator at −80 °C until RNA was extracted.

RNA extraction and qRT-PCR analysis

Total RNA was extracted by TRIzol reagent (Invitrogen, CA, USA) from tissues. After dissolution, NanoDrop Lite spectrophotometer to evaluate the concentration and purity of total RNA. Then, the total RNA reverse transcription synthesis of cDNA and used to conduct qRT-PCR assays for genes. GAPDH was used as internal control, and PCR experiments were repeated three times. Fold change (2−ΔΔCT) was represented relative gene expression levels. Table 1 was listed all the primer sequences and small interfernce sequences, and divergent primer was designed for each gene.

Table 1 mRNA PCR primer.

Gene name	Primer sequence	
FN1	F: CGGTGGCTGTCAGTCAAAG	
R: AAACCTCGGCTTCCTCCATAA	
CCND1	F: GCTGCGAAGTGGAAACCATC	
R: CCTCCTTCTGCACACATTTGAA	
CDH2	F: TCAGGCGTCTGTAGAGGCTT	
R: ATGCACATCCTTCGATAAGACTG	
CXCL12	F: ATTCTCAACACTCCAAACTGTGC	
R: ACTTTAGCTTCGGGTCAATGC	
MET	F: AGCAATGGGGAGTGTAAAGAGG	
R: CCCAGTCTTGTACTCAGCAAC	
IRS1	F: ACAAACGCTTCTTCGTACTGC	
R: AGTCAGCCCGCTTGTTGATG	
DCN	F: ATGAAGGCCACTATCATCCTCC	
R: GTCGCGGTCATCAGGAACTT	
FMOD	F: GAGACCTACGAGCCTTACCC	
R: TTGAGGTTGCGATTGTCACAG	

Results

Intersection of DEGs in four GEO datasets

As shown in Fig. 1, we conducted a multi-step analysis to study hub DEGs and its important biological functions and prognosis in THCA through comprehensive bioinformatics methods. To begin with, we chose and obtained a total of four gene expression profiles of THCA from GEO datasets (GSE3467, GSE3678, GSE33630 and GSE53157). In this research, 100 cases of thyroid tumors and 64 cases of normal tissues were obtained.

The DEG distribution in every GEO datasets demonstrated in the form of volcano plots (Figs. 2A–2D), including 851 DEGs in GSE3467, 785 DEGs in GSE3678, 1404 DEGs in GSE33630, and 858 DEGs in GSE53157. The intersection of 179 DEGs in four GEO datasets was displayed in Fig. 2E.

Functional annotation of DEGs

GO enrichment analysis was carried out to research the biological functions of all the DEGs. The BP classification of GO analysis demonstrated that the DEGs obviously enriched the skeletal system development, muscle cell differentiation and regulation of chromosome organization. For CC, these DEGs are rich in chromatin, adherens junction and nuclear chromosome part. In addition they are significantly rich in transcription coregulator activity, proximal promoter sequence-specific DNA binding and RNA polymerase II proximal from promoter in the MF category (Fig. 3A).

Figure 1 The development and verification process of the assay method is summarized and detailed in the following sections.

Gene Expression Omnibus; DEG, differentially expressed genes; PPI, protein–protein interaction; TCGA, The Cancer Genome Atlas.

Figure 2 Filter and identify DEGs in various GEO datasets.

(A–D) The volcanic map of the DEGs distribution was shown in GSE3467 (A), GSE3678 (B), GSE33630 (C) and GSE53157 (D). (E) Intersection of four datasets. DEG, differentially expressed gene.

Figure 3 Gene Ontology and Kyoto Encyclopaedia of Genes and Genomes pathway enrichment analyses.

(A) Genes were assigned to GO categories and the terms were summarized into three main GO categories. (B) Enrichment Analysis of abnormal regulation DEGs in Bubble Diagram. DEG, differentially expressed gene.

The enrichment analysis of KEGG pathway displayed that DEGs were obviously abundant in metabolic pathways, TGF-beta signaling pathway, central carbon metabolism in cancer and proteoglycans in cancer (Fig. 3B). Pathways of PTC were also contained, including metabolic pathways, neuroactive ligand-receptor interaction, rap1 signaling pathway, thyroid hormone signaling pathway.

Construction of PPI network

In order to determine the interaction of these DEGs in THCA, we used the STRING database to build a PPI network in Cytoscape software (Fig. 4A), which included 127 nodes and 199 edges. Next, we use the plug-in cytoHubba in Cytoscape to screen out the first eight hub genes. They are FN1, CCND1, CDH2, CXCL12, MET, IRS1, DCN and FMOD using the MCC method in Fig. 4B.

Figure 4 PPI of DEGs.

(A) The red nodes represent the hub genes. (B) A reader node means a higher degree.

Prognostic gene screening

In order to verify above results of all the DEGs and candidate genes obtained in other datasets, we obtained the gene expression profiles of all 179 DEGs in THCA from the TCGA database. Additionally, a distinctly different expression pattern between the tumor and normal samples were found (Fig. 5A–5H). We explored the association between gene expression and survival, and found that only DCN was significantly correlated with the overall survival (OS) of THCA samples trough the GEPIA database (log-rank p = 0.01892, Fig. 5I).

Figure 5 The expression of genes on TCGA.

(A) FN1. (B) CCND1 (C) CDH2 (D) CXCL12 (E) MET (F) IRS1 (G) DCN (H) FMOD (I). The OS of DCN in TCGA database. Red represent cancer tissues, grey represent normal tissues. * means P < 0.05. TCGA, The Cancer Genome Atlas; OS, overall survival.

Validation of differentially expressed levels for hub genes by qRT-PCR

The levels of eight hub genes expression (FN1, CCND1, CDH2, CXCL12, MET, IRS1, DCN and FMOD) in PTC and para-cancerous non-tumor tissues were detected by qRT-PCR. As shown in Fig. 6, FN1 (fibronectin 1), CCND1 (Cyclin D1), CDH2 (cadherin 2), MET (MET proto-oncogene, receptor tyrosine kinase), were significantly upregulated in THCA tissues (n = 87) compared with normal samples. Additionally, IRS1 (insulin receptor substrate 1), DCN (decorin), FMOD (fibromodulin), and CXCL12 (chemokine C-X-C motif ligand 12) were down-regulated in tumor tissues. The above results confirmed the accuracy of our analysis and showed that these eight key genes can be considered as a potential signature or biomarker of THCA.

Figure 6 Relative expression levels of hub genes detected by qRT-PCR.

* p < 0.05, ** p < 0.01 and *** p < 0.001. (A) FN1. (B) CCND1 (C) CDH2 (D) CXCL12 (E) MET (F) IRS1 (G) DCN (H) FMOD. qRT-PCR, quantitative real time polymerase chain reaction.

Discussion

In the present study, we screened 179 robust DEGs in PTC’s four GEO datasets. These DEGs were obviously enriched in a variety of cancer-related functions and pathways by GO and KEGG pathway enrichment analysis. Use the STRING database to build the PPI network, and carry on the hub genes. We finally screened eight central genes from the whole network, including FN1, CCND1, CDH2, CXCL12, MET, IRS1, DCN and FMOD, which genes are involved in the regulation of cyclin gene or cadherin gene, and could participate in the development of PTC. The expression verification and prognosis analysis of these candidate genes depended on TCGA database showed the robustness and reliability of the above results.

Several studies had been published about DEGs in human malignant tumors, such as THCA. Liu et al. (2019) identified 359 DEGs between tumor and normal thyroid tissues from TCGA database. Among them, five DEGs (LPAR5, NMU, FN1, NPY1R, and CXCL12) were screened with higher degrees and LPAR5 was associated with OS. Compared with our research, among the top eight hub genes (FN1, CCND1, CDH2, CXCL12, MET, IRS1, DCN and FMOD), we found that only DCN existed significance of OS in this study.

Compared with previously published researches on THCA, our research fully integrates multiple databases, datasets, and large sample sizes, including tumors and normal tissues. The results we got were robust and powerful, which could have non-invasive examination for patients with PTC.

Through palpation, doctors can find thyroid nodules, which are hard in texture and not smooth on the surface. At the same time, if the nodules do not move with the trachea when swallowing, and the enlarged and hard lymph nodes in the neck can be touched, thyroid cancer should be taken into account. If the tumor is in a small stage, ultrasound has become a very good screening tool for thyroid space-occupying lesions, which is widely used in thyroid physical examination and clinical differential diagnosis. However, the ultrasonic features of benign and malignant nodules overlap, and ultrasound doctors’ interpretations and suggestions on these features are quite different in clinical practice, which requires further fine needle puncture and genetic testing at the same time. We obtained eight hub genes in the network, which can be used for unique expression levels in PTC, and we can even diagnose PTC in early diagnosis. Some of which have been proved to be associated with PTC, depended on PPI network and candidate genes evaluation. Low fibronectin 1 (FN1) expression in tumorous tissues was an independent worse prognostic factor for progression-free survival in sporadic medullary thyroid carcinoma (Zhan et al., 2018). Additionally, Ouchi et al. identified FN1 as a fusion partner of ALK by 5′RACE, FN1-ALK, resulted in ALK overexpression in the inflammatory myofibroblastic tumors, and this finding should facilitate development of novel therapeutics (Ouchi et al., 2015). Also, FN1 could encode fibronectin, which was associated with various cancers precession, including cervical cancer (Wang et al., 2019a), gastric cancer (Jiang et al., 2019), colorectal cancer (Wu et al., 2016). Cyclin D1 (CCND1) functions as regulators of CDK kinases, and forms a complex with and functions as a regulatory subunit of CDK4 or CDK6, whose activity is required for cell cycle G1/S transition. It was proven as a DEG and used as a diagnostic and predictive biomarker in thyroid carcinoma and may be functionally involved in the development and progression of the disease (Jeon et al., 2018). In addition, down regulation of CCND1 may be an important mechanism by which everolimus increases the therapeutic window of paclitaxel in cervical cancers (Yilmaz et al., 2016). The biological role of cadherin 2 (CDH2) was also mostly reported in glioma with prognostic significance (Chen, Cai & Jiang, 2018). Previous studies showed that chemokine (C-X-C motif) ligand 12 (CXCL12) had reported in different human tumors. CXCL12 could activate CXCR4 as well as multiple downstream multiple tumorigenic signaling pathways, promoting expression of various oncogenes. Activation of the CXCL12-CXCR4 signaling axis promotes epithelial-mesenchymal transition (EMT) (Sleightholm et al., 2017; Yang, Hu & Zhou, 2019). The biological roles of CXCL12 also were founded in THCA, in which CXCR4/CXCR7/CXCL12 axis offer new valuable insight into the oncogenesis of metastatic follicular thyroid carcinoma (Werner et al., 2018). The amplification of MET proto-oncogene, receptor tyrosine kinase (MET) is confirmed as a resistance factor in gastric cancer cells (Ebert et al., 2019). And, MET may serve a role in regulating PD-L1 expression in hepatocellular carcinoma (Chun & Hong, 2019) Meanwhile, MET is also involved in high-risk metastasis progression in thyroid cancer (Garcia et al., 2019). Insulin receptor substrate 1 (IRS1) can be phosphorylated to activate AKT kinase, which promotes the development of lung cancer (Gorgisen et al., 2019). The low expression of decorin (DCN) has been demonstrated to contribute to be as a potential biomarker in colon cancer (Li et al., 2017). DCN was found underexpressed in thyroid tumors (Arnaldi et al., 2005). Qian et al. (2014) had shown that the methylated +58CpG in DCN 5′-UTR involved in reduced the mRNA expression of DCN. In our research, Fibromodulin (FMOD) acted as an important mediator in VEGF expression and angiogenesis induced by GDNF in human glioblastoma (Chen et al., 2018).

The expression of DCN gene in cancer tissues was lower than that in para-cancerous tissues. However, the lower the expression in cancer tissue, the better the prognosis was. This DCN is still an oncogene, but its pathway may be suppressed as a whole, with low expression or low levels of other regulation. It can still have an effect on tumor growth, but it is not the main driving gene. This can also happen if the prognosis of the tumor is good.

However, functional experiments should be carried out to verify our results. In addition, further research on a larger sample size will be required.

Conclusion

We found eight hub genes including four up-regulated and four down-regulated in PTC, and disclosed that DCN gene is an important gene related to prognosis through TCGA database to verify the accuracy of our finding. This hub genes may be considered as a new and potential signature or biomarker of PTC, which will further aid in contributing to the diagnosis, prognosis and clinical application of therapeutic targets for PTC.

Supplemental Information

Supplemental Information 1 Raw data of qRT-PCR

Click here for additional data file.

Additional Information and Declarations

Competing Interests

Author Contributions

Human Ethics

Data Availability

The authors declare there are no competing interests.

Ying Wan conceived and designed the experiments, performed the experiments, prepared figures and/or tables, authored or reviewed drafts of the paper, and approved the final draft.

Xiaolian Zhang performed the experiments, authored or reviewed drafts of the paper, and approved the final draft.

Huilin Leng analyzed the data, prepared figures and/or tables, and approved the final draft.

Weihua Yin and Congling Zhang analyzed the data, authored or reviewed drafts of the paper, and approved the final draft.

Wenxing Zeng performed the experiments, prepared figures and/or tables, and approved the final draft.

The following information was supplied relating to ethical approvals (i.e., approving body and any reference numbers):

The Ethical Committee of People’s Hospital of Yichun City approved this study (PHYC-2018-10-8-1).

The following information was supplied regarding data availability:

Data is available at NCBI GEO: GSE3467, GSE3678, GSE33630 and GSE53157 and from the TCGA database using search terms TCGA-THCA.

PCR data is available as Supplemental File.

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
