# Peer review of "Identifying hub genes of papillary thyroid carcinoma in the TCGA and GEO database using bioinformatics analysis"

_PeerJ, doi:10.7717/peerj.9120_

## Round 0.1 · original submission · Major Revisions

Please address critical issues raised by the reviewers and revise your manuscript accordingly.

Reviewer 1 ·

Basic reporting

Article fails to meet our standards in basic reporting and no novelty in this work

Experimental design

article fails to meet our standards in experimental design

Validity of the findings

Article fails to meet our standards in validity of the findings

Additional comments

Author performed reanalyzed work.

GEO accession no GSE3467 was previously analysed, work done and published.
List of of following publication (titles) along with URL about GEO accession no GSE3467

1. https://www.spandidos-publications.com/or/40/1/111 (Integrated bioinformatics analysis reveals that the expression of cathepsin S is associated with lymph node metastasis and poor prognosis in papillary thyroid cancer)

2. https://www.spandidos-publications.com/mmr/14/6/5041 (Identification of potential biomarkers and drugs for papillary thyroid cancer based on gene expression profile analysis)

3. https://portlandpress.com/bioscirep/article/39/4/BSR20190083/110840/Identification-and-analysis-of-genes-associated (Identification and analysis of genes associated with papillary thyroid carcinoma by bioinformatics methods)

4. https://link.springer.com/article/10.1007/s40618-018-0859-3 (Identification of key genes of papillary thyroid cancer using integrated bioinformatics analysis)

5. https://peerj.com/articles/7441/?utm_source=TrendMD&utm_campaign=PeerJ_TrendMD_1&utm_medium=TrendMD (Identification of pivotal lncRNAs in papillary thyroid cancer using lncRNA–mRNA–miRNA ceRNA network analysis)

6. https://wjso.biomedcentral.com/articles/10.1186/s12957-017-1190-8 (Special role of JUN in papillary thyroid carcinoma based on bioinformatics analysis)

7. https://link.springer.com/article/10.1007/s12253-013-9625-1 (Candidate Agents for Papillary Thyroid Cancer Identified by Gene Expression Analysis)

8. https://www.ncbi.nlm.nih.gov/pmc/articles/PMC6182223/ (SDC4 Gene Silencing Favors Human Papillary Thyroid Carcinoma Cell Apoptosis and Inhibits Epithelial Mesenchymal Transition via Wnt/β-Catenin Pathway)

9. https://onlinelibrary.wiley.com/doi/abs/10.1002/jcp.28932 (Identification of key genes and pathways of thyroid cancer by integrated bioinformatics analysis)

10. https://onlinelibrary.wiley.com/doi/abs/10.1002/jcp.29154 (microRNA‐599 promotes apoptosis and represses proliferation and epithelial‐mesenchymal transition of papillary thyroid carcinoma cells via downregulation of Hey2‐depentent Notch signaling pathway)

GEO accession no GSE3678 was previously analysed, work done and published.
List of of following publication (titles) along with URL about GEO accession no GSE3678

1. https://www.sciencedirect.com/science/article/pii/S0014299918306812 (Identification and validation of potential target genes in papillary thyroid cancer)

2. https://www.spandidos-publications.com/mmr/14/6/5041 (Identification of potential biomarkers and drugs for papillary thyroid cancer based on gene expression profile analysis)

3. https://www.spandidos-publications.com/10.3892/etm.2018.6023 (Identification and bioinformatics analysis of overlapping differentially expressed genes in depression, papillary thyroid cancer and uterine fibroids)

4. https://www.spandidos-publications.com/ijo/51/4/1311?text=fulltext (Anoctamin5 regulates cell migration and invasion in thyroid cancer)

5. https://www.ncbi.nlm.nih.gov/pmc/articles/PMC5650329/ (RNA-sequencing investigation identifies an effective risk score generated by three novel lncRNAs for the survival of papillary thyroid cancer patients)

6. https://www.thieme-connect.com/products/ejournals/html/10.1055/s-0035-1569289 (Identification of Genes Associated with Papillary Thyroid Carcinoma (PTC) for Diagnosis by Integrated Analysis)

7. https://journals.plos.org/plosone/article?id=10.1371/journal.pone.0197007 (Preserved SCN4B expression is an independent indicator of favorable recurrence-free survival in classical papillary thyroid cancer)

8. https://onlinelibrary.wiley.com/doi/full/10.1002/ijc.29172 (A three‐gene panel that distinguishes benign from malignant thyroid nodules)

9. https://peerj.com/articles/7441/?utm_source=TrendMD&utm_campaign=PeerJ_TrendMD_1&utm_medium=TrendMD (Identification of pivotal lncRNAs in papillary thyroid cancer using lncRNA–mRNA–miRNA ceRNA network analysis)

10. https://wjso.biomedcentral.com/articles/10.1186/s12957-017-1190-8 (Special role of JUN in papillary thyroid carcinoma based on bioinformatics analysis)

GEO accession no GSE33630 was previously analysed, work done and published.
List of of following publication (titles) along with URL about GEO accession no GSE33630

1. https://www.spandidos-publications.com/or/40/1/111 (Integrated bioinformatics analysis reveals that the expression of cathepsin S is associated with lymph node metastasis and poor prognosis in papillary thyroid cancer)

2. https://www.spandidos-publications.com/or/36/5/3005 (Elucidation of the molecular mechanisms of anaplastic thyroid carcinoma by integrated miRNA and mRNA analysis)

3. https://www.spandidos-publications.com/mmr/18/1/695?text=fulltext (Competing endogenous RNA regulatory network in papillary thyroid carcinoma)

4. https://www.sciencedirect.com/science/article/pii/S0014299918306812 (Identification and validation of potential target genes in papillary thyroid cancer)

5. https://portlandpress.com/bioscirep/article/39/4/BSR20181616/110809/Down-regulation-of-DANCR-acts-as-a-potential (Down-regulation of DANCR acts as a potential biomarker for papillary thyroid cancer diagnosis)

6. https://www.ncbi.nlm.nih.gov/pmc/articles/PMC6492601/ (Long Non-Coding RNA ZFAS1 as a Novel Potential Biomarker for Predicting the Prognosis of Thyroid Cancer)

7. https://link.springer.com/article/10.1007/s12253-018-0561-y (Down-Regulation of APTR and it’s Diagnostic Value in Papillary and Anaplastic Thyroid Cancer)

8. https://www.futuremedicine.com/doi/abs/10.2217/fon-2019-0016 (LncRNAs SNHG12 and LINC00152 were associated with progression of patients with papillary thyroid carcinoma)

9. https://www.thieme-connect.com/products/ejournals/html/10.1055/s-0035-1569289 (Identification of Genes Associated with Papillary Thyroid Carcinoma (PTC) for Diagnosis by Integrated Analysis)

10. https://www.spandidos-publications.com/ol/18/5/4726 (Long non‑coding RNA small nucleolar RNA host gene 7 is upregulated and promotes cell proliferation in thyroid cancer)

GEO accession no GSE53157 was previously analysed, work done and published.
List of of following publication (titles) along with URL about GEO accession no GSE53157

1. https://www.thieme-connect.com/products/ejournals/html/10.1055/s-0035-1569289 (Identification of Genes Associated with Papillary Thyroid Carcinoma (PTC) for Diagnosis by Integrated Analysis)

2. https://onlinelibrary.wiley.com/doi/full/10.1111/1759-7714.13270 (Integrated analysis identifies DUSP5 as a novel prognostic indicator for thyroid follicular carcinoma)

3. https://link.springer.com/article/10.1007/s00418-018-1660-2 (Expression of serine peptidase inhibitor Kunitz type 1 in differentiated thyroid cancer)

4. https://onlinelibrary.wiley.com/doi/abs/10.1002/jcp.28793 (Integrated analysis of transcriptome data revealed MMP3 and MMP13 as critical genes in anaplastic thyroid cancer progression)

5. https://onlinelibrary.wiley.com/doi/abs/10.1002/jcp.28932 (Identification of key genes and pathways of thyroid cancer by integrated bioinformatics analysis)

6. https://onlinelibrary.wiley.com/doi/full/10.1002/cam4.1397 (Development and validation of an individualized diagnostic signature in thyroid cancer)

7. https://www.spandidos-publications.com/10.3892/ol.2015.3829 (Identification of potential therapeutic targets for papillary thyroid carcinoma by bioinformatics analysis)

8. https://link.springer.com/article/10.1007/s10238-016-0445-y (Overexpression of teneurin transmembrane protein 1 is a potential marker of disease progression in papillary thyroid carcinoma)

9. https://www.sciencedirect.com/science/article/abs/pii/S0378111918303470 (Network-based meta-analysis in the identification of biomarkers for papillary thyroid cancer)

10. https://www.spandidos-publications.com/10.3892/ol.2018.9342 (Genetic expression profile‑based screening of genes and pathways associated with papillary thyroid carcinoma)

Re analysed work is not acceptable
Mean while, Author not provided Differential gene expression table, which is more fundamental and basic in this work.
No novelty in this work.
I recommended strong rejection.

Reviewer 2 ·

Basic reporting

NO comment

Experimental design

No comment

Validity of the findings

No comment

Additional comments

The manuscript entitled “Identification of hub genes of thyroid carcinoma from TCGA and GEO database by bioinformatics analysis” by Ying Wan and co-workers reports eight hub genes of thyroid carcinoma identified using bioinformatic methods and verified using clinic human THCA patients tissue samples. The authors did a good job to do bioinformatic analysis to identify the eight hub genes from four expression profiles datasets, which include thyroid tumors and normal tissues. However, there are still some issues with this manuscript.
Major concerns:
1). The authors' conclusion is not supported by the evidence provided in the manuscript. The authors tried to identify genes that could serve as the signature and potential prognostic markers of THCA. However, the eight hub genes identified in the manuscript are not unique features of THCA, but also for many other cancers. The expression patterns of those eight genes are the same in human thyroid carcinoma patients and the TCGA database (figure 5 and figure 6), which also indicate those eight genes are generally altered in many cancer types. So, the conclusion would be any single of the eight hub genes or the combination of all eight genes can not be used as biomarkers of thyroid carcinoma, though they are differentially expressed in thyroid carcinoma patients.
Minor concerns:
1). The methods should include more details. For example, the GEO datasets the authors used in the manuscript should make a clear note about the sub-types of thyroid cancer (since there are four different types) of all samples, the type of normal tissues, and also patients' genders that those samples originated. That information can help readers to determine if the current analysis has a bias in certain respects; How the authors get the “Survival” plot in figure 5i; RNA extraction and qRT-PCR analysis should revise carefully. How the samples were extracted using TRIzol, what the authors mean of all procedures were repeated three times (technique repeats?)
2. The results also can include more details. For example, where are the eight hub genes the authors identified in the manuscript located in the GO analysis, the KEGG pathway, and what are the known functions of those genes? The authors mentioned THCA pathways but did not include any details about those pathways. It would be very helpful to include more details about THCA pathways since they are directly related to the topics in the manuscript.
3. In the discussion part, the authors should emphasize how those eight genes play specific roles in thyroid cancers compared to other cancer types. In other words, why those genes can be used in the diagnosis and prognosis of THCA, but not other cancer types. The authors should explain the specificity here.
4). There are some typos in the manuscript which need to correct. For example, page 7, line 99, “dynthesis” should correct to “synthesis”; page 7, line 112, “The volcano plots of The DEG……in the form of a volcano plots” should correct to “ The DEG……in the form of volcano plots”. There are still many other places, the authors need to revise carefully.

Reviewer 3 ·

Basic reporting

no comment

Experimental design

no comment

Validity of the findings

no comment

Additional comments

The authors have identified 8 HUB genes for thyroid carcinoma using expression profiles from the GEO database.

The experimental design is rigorous and the conclusions are clearly stated.

The result section of the manuscript feels very short. Could this section be expanded?

The discussion section lists the identified genes and their background literature robustly. However, all these HUB genes have been reported to be involved in other types of cancers. Can the authors find any specific and unique connection between these HUB genes and THCA?

The figure legends should have sufficient information to understand the respective figures. For example, Fig 5 doesn't mention which one is normal tissue and which one is cancer tissue (red vs grey). Fig 5i also have hard to read words inside the graph. Fig 5a-h also failed to mention the meaning and importance of the red stars. The entire paper depends on figure 5 (and 6) for validation and therefore more attention should be given to them.

The findings of this study are very clear: the authors have identified 8 HUB gens and concluded that one of them,DCN, is a key gene for TCGA prognosis. What this paper is missing is a specific discussion on how these hubs genes are potentially involved in thyroid cancer. It is understandable that a cyclin gene or a cadherin gene (etc.) would be differentially regulated in tumor tissues. The authors should point out why do they think that these 8 HUB gens are specific for THCA and not just a bunch of genes gets differentially expressed in any tumor tissue (which this paper very robustly reviewed in the discussion section).

I think in studies like this, it is important to validate some of the negative results. This can be done very easily using RT-PCR and by picking a couple of genes that didn't get enriched but are related to the HUB genes (like a different cyclin gene as a control for Cyclin D1 and so on). Any thought?

---

## Round 0.2 · Major Revisions

Please address the remaining critiques of the reviewers and amend your manuscript accordingly.

Reviewer 2 ·

Basic reporting

No comment

Experimental design

NO comment

Validity of the findings

No comment

Additional comments

Thanks for the authors to provide additonal information and answers my concerns. In general, the authors did a good job to identify the 8 hub genes. Since the aims of the manuscript is to apply to clinical diagnosis. I still have two concerns.
1). The authors said the 8 hub genes can be used to early diagnosis in the manuscript. However, I do not see any evidence provided by the authors. How does the authors came up with this conclusion? At least, the authors should address it more clear in the manuscript.
2). The authors verified the gene expression pattern of the 8 hub genes using two methods. One is bioinformatically analyzed the THCA datasets, the other way is to run qRT-PCR for the patients. I am convinced by the overall pattern changes of the 8 genes in throid cancer patients between cancer tissues and normal tissues. However, if look into details about the q-PCR results. I found in each gene group, there is about 15-30% patients did not show the overall trend change in the gene expression level (this even does not count cancer and normal tissue shows similar level of gene expression). For example, the overall pattern of gene FN1 is upregulated in cancer tissue, however, sample 7,13, 17, 22, 31, 47, 48 show FN1 is more abandunt in normal tissue. If the same sample ID for the q-PCR results for each of the 8 genes is for the same patient (for example, sample No.1 is patient No. 1). Then, the percentage of patients which show all trend changes of the 8 hub genes will be extrememly low.That means, even a throid cancer patient will not trypically show all 8 genes expression change, the authors should change their conclusions accordingly. The authors should analyze the data based on individual patient, and report the result in a way that a paired comparing between the cancer and normal tissue for the same patient is presented, at least for the q-PCR. Otherwise, the results will be misleading for a diagnosis purpose.

Reviewer 3 ·

Basic reporting

n/a

Experimental design

n/a

Validity of the findings

n/a

Additional comments

The argument of using all eight HUB genes together as a 'whole signature' is unconvincing. Especially when a significant number of the eight HUB genes are known to be associated with tumors and are central to various cellular growth functions. I agree the idea is plausible but the authors need to use proper data and stronger arguments.

I think the responses to the first reviewer's comments are extremely poor. I agree that the reanalysis of data may be acceptable under specific scenarios. However, the authors only stated that they disagree with reviewer 1, but didn't provide specific rebuttals to the reviewer's concerns. As the reviewer specifically cited various studies, it can be expected that the authors would use this opportunity to provide proper documentation on why they believe their work contains novel findings especially under the lights of the mentioned papers.

I recommend the authors to go through the major concerns pointed out by the reviewers and try to argue on specific points.

---

## Round 0.3 · Major Revisions

Please address the remaining critiques of the reviewers and amend your manuscript accordingly.

Reviewer 2 ·

Basic reporting

no comment

Experimental design

no comment

Validity of the findings

no comment

Additional comments

I am generally agreed to publish the manuscript as is. However, I still encourage the authors to present the raw rt-pcr data to represent each individual patient in addition to the statistic analysis. To do this, just lable the raw data like patient No.1 corresponding to each rt-pcr data for the fisrt patient, and patient No.2 to the second patient rt-pcr data, etc.

Reviewer 3 ·

Basic reporting

.

Experimental design

.

Validity of the findings

.

Additional comments

The rebuttal letter doesn't contain any response to Reviewer 3 (wrong file? uploading error?). Please go through the previous comments and respond to them accordingly.

Previous Comments:
The argument of using all eight HUB genes together as a 'whole signature' is unconvincing. Especially when a significant number of the eight HUB genes are known to be associated with tumors and are central to various cellular growth functions. I agree the idea is plausible but the authors need to use proper data and stronger arguments.

I think the responses to the first reviewer's comments are extremely poor. I agree that the reanalysis of data may be acceptable under specific scenarios. However, the authors only stated that they disagree with reviewer 1, but didn't provide specific rebuttals to the reviewer's concerns. As the reviewer specifically cited various studies, it can be expected that the authors would use this opportunity to provide proper documentation on why they believe their work contains novel findings especially under the lights of the mentioned papers.

I recommend the authors to go through the major concerns pointed out by the reviewers and try to argue on specific points.

---

## Round 0.4 · Minor Revisions

Please address remaining issues pointed by the reviewers.

Reviewer 2 ·

Basic reporting

No Comment

Experimental design

No Comment

Validity of the findings

No Comment

Additional comments

I still encourage or request the authors to proper present the clinlical q-PCR data. In a way that the expression levels of the 8 genes for each patient are clearly labeled in this study. The authors did not provide strong argument that the combination of the 8 hub gene expression patterns is proper in thyroid cancer diagnosis. I appreciate the authors effort to try to develop a new diagnosis tool for the thyroid cancer. However, the authros have to clealy and properly report their findings. The statistic significant changes in genes expressions between the cancer tissues and normal tissues does not mean for a invidual patient you will see that change. Especially, the authors did not use paired statistic analysis to link every data point to a specific patient. That's why the minimum requirement for the authors would be proper label their q-PCR data to let future readers to have an idea how robust this diagnosis method will be and how to improve it in the future.

Reviewer 3 ·

Basic reporting

.

Experimental design

.

Validity of the findings

.

Additional comments

I am happy with most of the responses the authors provided. Some of the conclusions are little stretched and could be toned down in the final proof. I still didn't get a proper reply in support of the novelty of the study, but I would leave that to the editor to decide (especially under the present lockdown scenarios).

---

## Round 0.5 · accepted · Accept

Since all the remaining critiques were adequately addressed and the manuscript was amended accordingly, I am pleased to accept the revised version in its current form.